# On the 3D → 2D Isomerization of Hexaborane(12)

**Josep M. Oliva-Enrich** [1,*], **Ibon Alkorta** [2], **José Elguero** [2], **Maxime Ferrer** [2] and **José I. Burgos** [3]

1    Instituto de Química-Física "Rocasolano" (CSIC), Serrano 119, E-28006 Madrid, Spain
2    Instituto de Quimica Médica (CSIC), Juan de la Cierva, 3, E-28006 Madrid, Spain; ibon@iqm.csic.es (I.A.); iqmbe17@iqm.csic.es (J.E.); maxime.ferrer2802@gmail.com (M.F.)
3    Instituto de Ciencias Matemáticas (CSIC-UAM-UCM-UC3), Nicolás Cabrera 15, E-28049 Madrid, Spain; burgos@icmat.es
*    Correspondence: j.m.oliva@iqfr.csic.es; Tel.: +34-91-5619400

**Abstract:** By following the intrinsic reaction coordinate connecting transition states with energy minima on the potential energy surface, we have determined the reaction steps connecting three-dimensional hexaborane(12) with unknown planar two-dimensional hexaborane(12). In an effort to predict the potential synthesis of finite planar borane molecules, we found that the reaction limiting factor stems from the breaking of the central boron-boron bond perpendicular to the $C_2$ axis of rotation in three-dimensional hexaborane(12).

**Keywords:** boron; valence isomer; hexaborane(12); planarity; quantum chemistry; chemical bond

## 1. Introduction

The chemistry of boron is usually treated as a separate chapter in inorganic chemistry textbooks and is considered as more diverse and complex than that of any other element in the periodic table, with the exception of carbon [1]. Boranes are compounds combining boron and hydrogen $B_nH_m$. For low molecular weights they are sensitive to air and moisture, toxic, and volatile [2], such as pentaborane(9) $B_5H_9$, which is very flammable and acutely toxic; however, they can also be stable solids which can be handled under ambient conditions, such as $B_{10}H_{14}$ and $B_{18}H_{22}$. The unique three-dimensional (3D) structural and bonding patterns of boranes confer to them a rich variety of architectural molecular constructs [3], and the combination of boranes with metals and other elements of the periodic table leads to compounds included in emerging fields of current fundamental and applied research [4].

On the other hand, two-dimensional (2D) planar borane molecules are unknown due to the 3D clusterization of boron atoms when forming many-electron multicenter bonds [5]. However, recent experiments [6] called our attention to the possibility of isolating planar finite borane molecules. Moreover, a one-to-one correspondence between any conjugated hydrocarbon $C_nH_m$ and the structurally equivalent borane $B_nH_{m+n}$ can be easily drawn [7]. For instance, benzene $C_6H_6$ can be transformed into planar hexaborane(12) $B_6H_{12}$ by substituting carbon atoms with boron atoms and every π two-electron bond with one perpendicular $H_2$ moiety at the mid-point of the former C=C bond; in other words, with three {C=C} → {BH$_2$B} substitutions [7,8]. Therefore, potential synthesis of planar borane molecules encompasses a new field of research within boron chemistry.

We know that *arachno*-$B_6H_{12}$, hexaborane(12), is a colorless liquid that, like most boron hydrides, is readily hydrolyzed and flammable, and usually prepared from $[B_5H_8]^-$, the conjugate base of pentaborane(9), $B_5H_9$ [9]. Derivatives of hexaborane(12) have been synthesized and characterized [10–12], and its thermal gas-phase decomposition has been studied [13]. This molecule has a 3D (curved) structure with $C_2$ symmetry, as shown in Figure 1a,b [14]. Thus, the 3D → 2D isomerization of hexaborane(12) to the unknown planar $D_{3h}$ structure, Figure 1c,d, corresponds to a flattening and swelling of the $B_6$ skeleton into a planar $B_6$ hexagon. According to Lipscomb's *styx* notation [15], there are four possible

isomers with the $B_6H_{12}$ formula: 6030, 5121, 4212, and 3303. In *styx* notation, *s* stands for number of bridge hydrogens, *t* for the number of two-electron three-center boron bonds, *y* for the number of two-center two-electron boron bonds, and *x* for the number of $BH_2$ groups. Reactant ($C_2$) and Product ($D_{3h}$) [7,8] in the 3D → 2D hexaborane(12) isomerization correspond to isomers 4212 and 6030, respectively, and structure 5121 to an intermediate of the 3D → 2D isomerization, as shown below. Isomer 3303 lies 13 kJ·mol$^{-1}$ above R (4212). Isomers 6030 (P) and 5121 lie 100 kJ·mol$^{-1}$ and 104 kJ·mol$^{-1}$ higher than R (4212). A summary of the structural description and relative energies of hexaborane(12) isomers with *styx* notation is included in the Supplementary Information, Table S1. The question that we would like to answer in this work, dedicated to Professor Josef Michl, is related to the possibility of transforming 3D hexaborane(12) into a 2D planar structure through chemical reaction steps. Synthesis of planar borane molecules mimicking planar conjugated hydrocarbons is certainly a scientific challenge. What follows is an attempt to give an acceptable answer to this question.

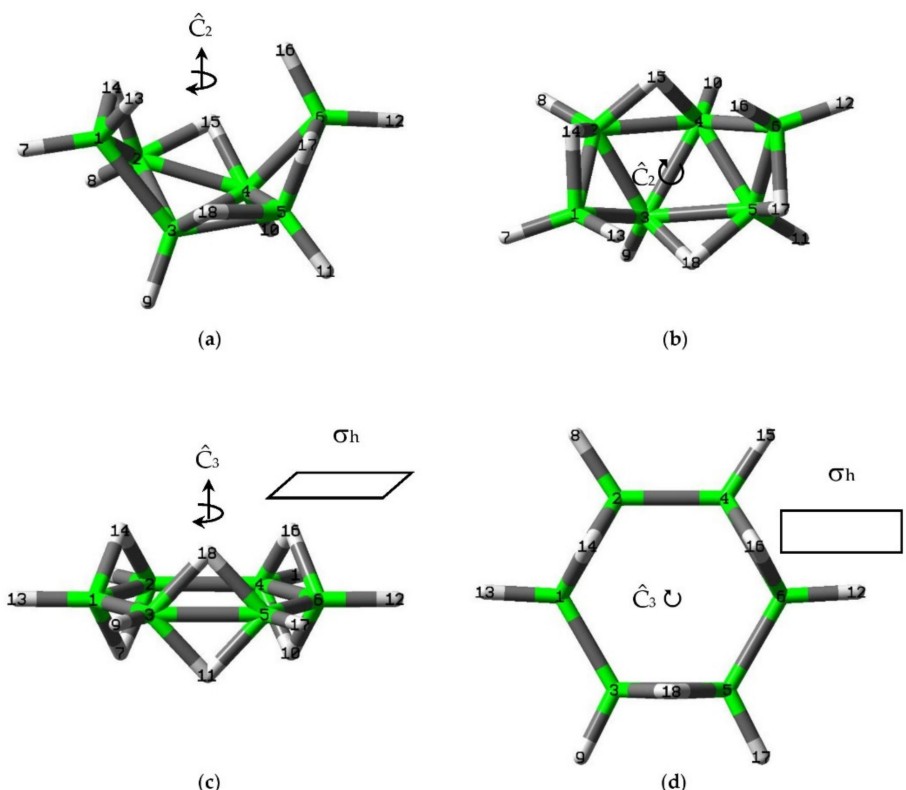

**Figure 1.** Two projections of known 3D hexaborane(12) with $C_2$ symmetry (**a**) perpendicular to the $\hat{C}_2$ rotation axis and (**b**) along the $\hat{C}_2$ rotation axis, and two projections of unknown planar 2D hexaborane(12) with $D_{3h}$ symmetry (**c**) perpendicular to the $\hat{C}_3$ rotation axis and $\sigma_h$ plane and (**d**) along the $\hat{C}_3$ rotation axis and on the $\sigma_h$ plane. Atom labels are shown for boron (green) and hydrogen (white).

## 2. Computational Methods

The electronic structure calculations presented in this work were carried out with the scientific software Gaussian16 [16] and the M06-2X/aug-cc-pVDZ level of theory [17–19], which consists of a hybrid functional combining density-functional and Hartree-Fock theory, which includes dispersion corrections, and an augmented double-$\zeta$ basis set with polarization and diffuse functions for all atoms. The transition states (TS), stationary points with one imaginary frequency, were located by combining synchronous transit and quasi-Newton methods [20,21]. In order to locate the intermediates (I) at either site of the TS col point we followed the vibrational mode of the imaginary frequency—forward and

backward—along the intrinsic reaction coordinate (IRC) [22,23] and relaxed the geometry for searching an energy (local) minimum. All TS and I were checked with frequency computations at the same level of theory, with one and zero imaginary frequencies, respectively. The quantum theory of Atoms-in-Molecules (QTAIM) [24,25] calculations were carried out with the AIMAll program [26]. In this theory, the electron density is analyzed from a topological point of view, with gradient and Laplacian operators, $\vec{\nabla}\rho$ and $\nabla^2\rho$, respectively. The critical points of the electron density are those with $\vec{\nabla}\rho_c = 0$; the three eigenvalues of $\nabla^2\rho_c$ are classified according to the number of non-zero values and the sum of the signs as follows: A maximum (3,−3) critical point is associated with nuclei positions, the saddle points (3,−1) and (3,+1) are associated with bond (BCP) and ring critical points (RCP), respectively, and the minima (3,+3) correspond to cage critical points. The molecules included in this work provide BCP and RCP, apart from the maxima corresponding to nuclear positions. A gradient path-line connects two nuclei through a BCP, where the electron density $\rho_{BCP}$ is a maximum in two directions and a minimum in one direction. In an RCP, the $\rho_{RCP}$ is a minimum in one direction and a maximum in two directions.

The molecular volume was computed with the Gaussian16 scientific software, as the volume inside a contour of $0.0067 \text{ e/Å}^3$ density, which is accurate to two significant figures, and carried out by Monte-Carlo integration.

## 3. Results

### 3.1. Intrinsic-Reaction-Coordinate (IRC) and Stationary Points in the 3D → 2D Isomerisation of Hexaborane(12)

In Figure 2 we plot the energy profile of the reaction pathway along the intrinsic reaction coordinate (IRC) from reactant (R), $C_2$ hexaborane(12), to product (P), $D_{3h}$ planar hexaborane(12), with the structures of transition states (TS) and intermediates (I). The nature of all stationary points (SP)—TS and I—along the reaction pathway were checked with frequency computations, with one and zero imaginary frequencies, respectively. Between R and P we found five stationary points on the energy hypersurface along the IRC, with three transition states, $TS_1$, $TS_2$, and $TS_3$, and two intermediates, $I_1$ and $I_2$. In Table 1 we gather the energies of these SP and the energy differences with respect to R, the lowest energy isomer.

**Table 1.** Energy (a.u.) and energy differences $\Delta E_R = E(SP) - E(R)$ (kJ·mol$^{-1}$) for the stationary points (SP) of the 3D → 2D isomerization in hexaborane(12). R = Reactant, SP = Stationary Point, TS = Transition State, I = Intermediate. M06-2X/aug-cc-pVDZ calculations.

| SP | E | $\Delta E_R$ |
|----|----|----|
| R | −156.20882555 | 0.0 |
| $TS_1$ | −156.18016535 | 75.2 |
| $I_1$ | −156.19687483 | 31.4 |
| $TS_2$ | −156.10584340 | 270.4 |
| $I_2$ | −156.16931585 | 103.7 |
| $TS_3$ | −156.12728434 | 214.1 |
| P | −156.17070415 | 100.1 |

As gathered in Table 1 and displayed in Figure 2, unknown planar hexaborane(12)—P—lies 100 kJ·mol$^{-1}$ above existing hexaborane(12)—R. The three energy barriers from (local) energy minima to TS along the IRC are 75 kJ·mol$^{-1}$, 239 kJ·mol$^{-1}$, and 110 kJ·mol$^{-1}$ for $TS_1$, TS2, and $TS_3$ respectively and the intermediates $I_1$ and $I_2$ lie 31 kJ·mol$^{-1}$ and 104 kJ·mol$^{-1}$ above R, respectively.

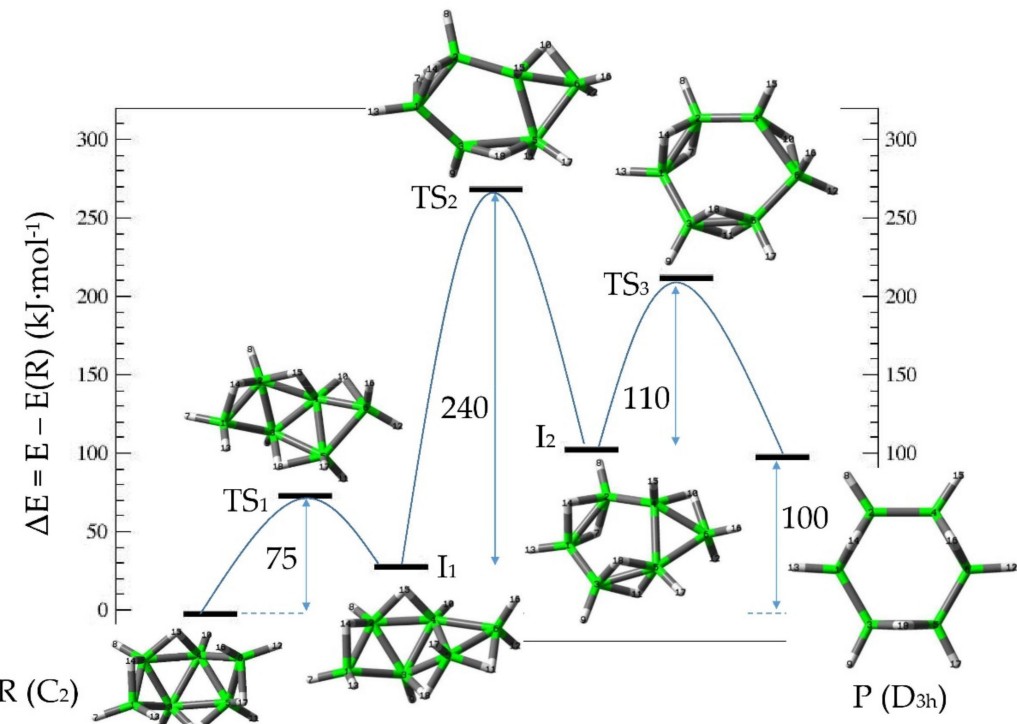

**Figure 2.** Reaction pathway along the IRC from reactant (R) 3D $C_2$ hexaborane(12) to product (P) 2D $D_{3h}$ hexaborane(12).

From a chemical point of view, as shown in Figure 2, the reaction determining step corresponds to the $I_1 \rightarrow TS_2$ process, with the largest barrier: 240 kJ·mol$^{-1}$. In this process the $B_2$-$B_3$ and $B_3$-$B_4$ bonds are broken, thus opening the central tilted $B_4$ rhombus. The first two steps, $R \rightarrow TS_1 \rightarrow I_1$ correspond to hydrogen bridge atoms moving so that the boron-frame structure rearranges, with an energy barrier of 75 kJ·mol$^{-1}$, comparable to energy barriers in $S_N2$ chemical reactions in organic chemistry [27]. After $TS_1$ we reach intermediate $I_2$, the predicted *styx* isomer 5121 (see Supplementary Information). As compared to R, the $I_2$ structure has one more bridge hydrogen atom, one less three-center B-B-B bond, an additional two-center B-B bond, and one less $BH_2$ group. From a chemical point of view $I_2$ has a similar energy as compared to P, with a difference of 3.6 kJ·mol$^{-1}$ only. The transition state $TS_3$, separating $I_2$ and P, lies 110 kJ·mol$^{-1}$ above $I_2$.

Given the complexity of the borane cage rearrangements along the IRC, we have selected boron-boron B(i)-B(j)—Figure 3—and boron-hydrogen B(i)-H(j) distances—Figure 4—in the SP, along the reaction coordinate from R to P. Thus, as displayed in Figure 1a,b, B(1) has the following connectivities: B(1)-B(2), B(2)-B(3), B(1)-H(7), B(1)-H(13), and B(1)-H(14). For the remaining atoms a similar reasoning is followed and in the Supplementary Information file (Supplementary Information) we provide the values for all the B(i)-B(j) and B(i)-H(j) distances for SP in Tables S2 and S3 respectively. In the Supplementary Information file we also provide the optimized geometries of all SP points included in this work, Tables S4–S11. In Figures 3 and 4 all B(i)-B(j) and selected B(i)-H(j) distances—in the latter those with major changes, respectively—are plotted along the IRC for the 3D → 2D isomerization of hexaborane(12).

In Figure 3 one can clearly envisage two subsets of B(*i*)-B(*j*) connectivities. We should take into account that in R, the equivalent B(1)-B(3) and B(4)-B(6) distance is 1.904 Å, and therefore the interaction between boron atoms is not so strong as compared to the stiffer central tilted rhombus formed by B(2), B(3), B(4), and B(5) with boron-boron distances B(2)-B(3) = B(4)-B(5) = 1.724 Å (red line in Figure 3), B(2)-B(4) = B(3)-B(5) = 1.798 Å (cyan line in Figure 3), and the central B(3)-B(4) = 1.786 Å (black line in Figure 3). In the first transition state $TS_1$, with an energy barrier of 75 kJ·mol$^{-1}$ from R, the B(4)-B(6) distance (grey line in Figure 3) decreases by 0.2 Å and the B(5)-B(6) distance increases by the same amount, while

the remaining B-B distances remain similar. However, as displayed in Figure 4, the B(6)-H(10) and B(6)-H(17) distances undergo significant changes with a decrease and increase of 1 Å, respectively. The remaining B-H distances remain almost unaltered along the R → TS$_1$ step. As we move from TS$_1$ to the first intermediate I$_1$, the B-B distances remain almost unaltered with the exception of B(4)-B(6) and B(5)-B(6) which seem to exchange their profiles, returning to the original values in R. As for the B-H distances, Figure 4, there is an increase of 0.3 Å for B(3)-H(11), B(4)-H(16), B(6)-H(10), and B(6)-H(17). For B(2)-H(7) and B(2)-H(15) there is a slight decrease and increase of 0.1 Å respectively. At this point we should emphasize that the largest energy barrier for the planarization of B$_6$H$_{12}$ corresponds to the reaction step I$_1$ → TS$_2$, as displayed in Figure 2, with an energy barrier of 240 kJ·mol$^{-1}$ from I$_1$. This is the reaction determining step in the isomerization process.

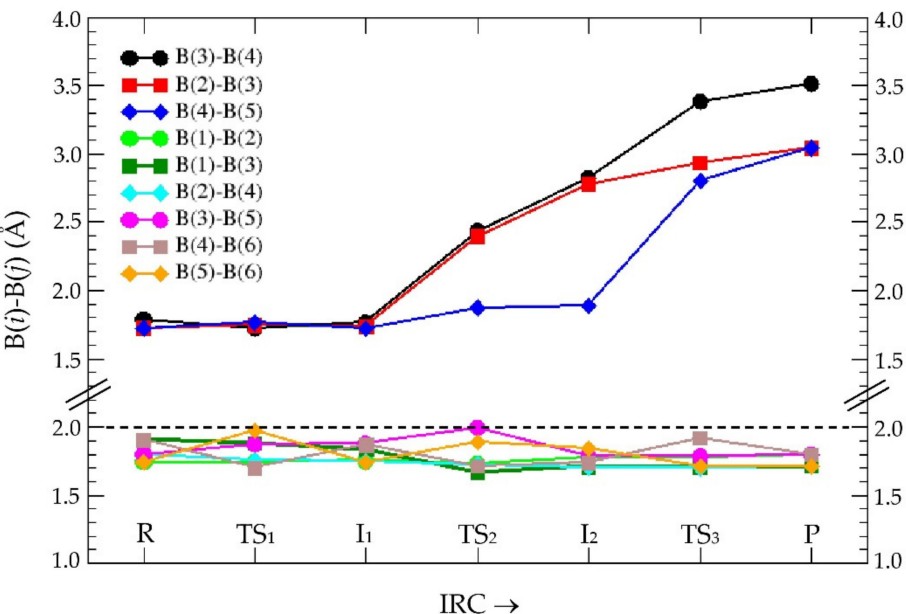

**Figure 3.** B($i$)-B($j$) distances (Å) for the Stationary Points (SP) along the R → P reaction pathway for the 3D → 2D isomerization of hexaborane(12).

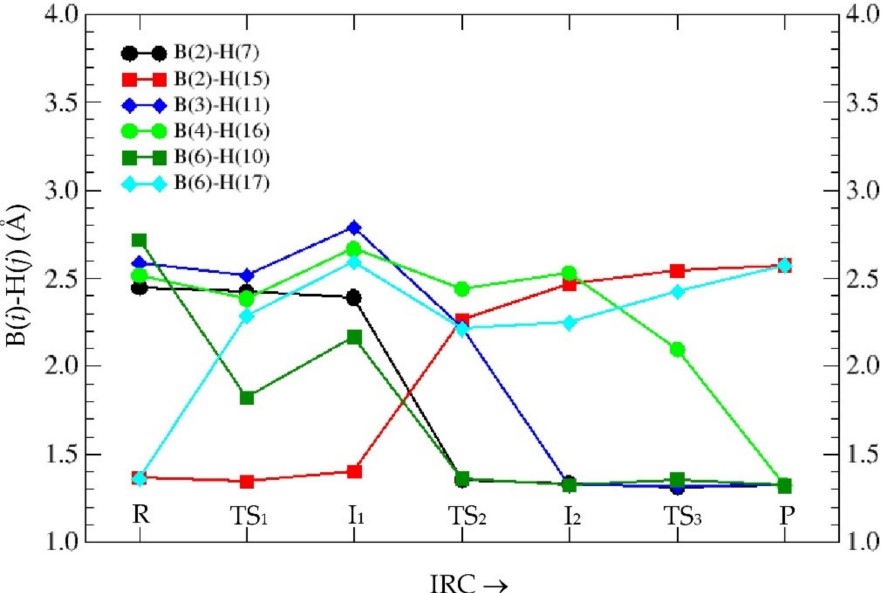

**Figure 4.** Selected B($i$)-H($j$) distances (Å) for the Stationary Points (SP) along the R → P reaction pathway for the 3D → 2D isomerization of hexaborane(12).

*3.2. QTAIM Analysis of the Stationary Points in the 3D → 2D Isomerisation of Hexaborane(12)*

The electron density of the SP along the IRC in the 3D → 2D isomerization of hexaborane(12) was also analyzed with the quantum theory of atoms in molecules (QTAIM), in order to further assess the geometrical and electronic structure changes along the IRC, as shown in Figure 5. QTAIM defines the chemical bonding and structure of a chemical system based on the topology of the electron density $\rho(\mathbf{r})$, with stationary (critical) points and gradient—bond—paths of the electron density that originate and terminate at these points. Thus, the critical points found between two atoms, called bond critical points (BCP), provide information about the nature of such bond. At this point we should emphasize that QTAIM (Figure 5) and IRC (Figure 2) stationary points correspond, respectively, to $\vec{\nabla}\rho = \left(\frac{\partial\rho}{\partial x}, \frac{\partial\rho}{\partial y}, \frac{\partial\rho}{\partial z}\right) = \vec{0}$ and $\vec{\nabla}E = \left(\frac{\partial E}{\partial X}, \frac{\partial E}{\partial Y}, \frac{\partial E}{\partial Z}\right) = \vec{0}$, where lower and capital $(x, y, z)$ Cartesian coordinates correspond to electrons and nuclei respectively, and they should not be confused. As shown in Figure 5, bond critical points (BCP) are depicted in yellow, ring critical points (RCP) in red, and the gradient bond-paths of the electron density are represented as black solid lines connecting atoms. For further details on the QTAIM calculations the reader is referred to Section 2 above. As shown in Figure 5, starting off with the upper left corner, R hexaborane(12) shows two equivalent RCP and several BCP between boron atoms and between boron and hydrogen atoms. The $\hat{C}_2$ rotation axis passes through the midpoint of the B(3)-B(4) bond path, where the BCP lies, and is perpendicular to this bond. The above statement on the low B(1)-B(3) = B(4)-B(6) interactions is confirmed with the absence of a gradient path connecting these nuclei. At this point we should emphasize that the threshold in the electron density of BCP for bond path plotting is 0.1 au: solid line for $\rho(BCP) > 0.1$, and dashed line for $\rho(BCP) < 0.1$. The topological analysis of R further shows two types of hydrogens: the bridge hydrogens {H(14), H(15)}, and the equivalent {H(17), H(18)} through $\hat{C}_2$ rotations, in which gradient paths split into two toward two neighbor boron atoms. For the remaining hydrogens, only one gradient path connects them to the neighbor boron atom. As we move from R to $TS_1$ with an energy barrier of 75 kJ·mol$^{-1}$, an additional RCP appears inside the {B(1)B(2)H(14)} triad and lies very close to the new BCP between B(1) and B(2) and to the BCP between B(1) and H(14). Additionally, on the other side of the molecule the B(6)-H(17) gradient path with its corresponding BCP vanishes and a new B(4)-B(6) gradient path appears with the corresponding BCP between B(4) and B(6). Following the IRC from $TS_1$ to $I_1$ down 44 kJ·mol$^{-1}$, the molecule seems to get back to the previous conformational structure of R but in fact this is not the case, with vanishing of the B(1)-B(2) and B(1)-H(14) bond paths with the subsequent vanishing of the RCP within the {B(1)B(2)H(14)} triad. This rearrangement brings the new B(1)-B(3) bond path with the respective BCP. On the other side of the molecule the B(4)-B(6) bond path vanishes and the new B(5)-B(6) bond path appears with the respective BCP. We now turn to the most energetic reaction step mechanism in the 3D → 2D isomerization of hexaborane(12), depicted with a thicker black arrow in Figure 5: an energy barrier of 240 kJ·mol$^{-1}$ with an interesting atomic rearrangement in the $I_1$ → $TS_2$ step. A close look at $TS_2$ clearly shows a bond breaking of B(2)-B(3), B(3)-B(4)—the central boron-boron bond in the initial $C_2$ structure—and B(2)-H(15). These changes imply the formation of the new bonds B(2)-B(4), B(1)-H(14), B(1)-H(7), and B(2)-H(7). The central frame of $TS_2$ shapes almost to a hexagonal structure with the central RCP and another RCP within the {B(1)H(7)H(14)}B(2) moiety resembling a diborane(6) structure. The next reaction step $TS_2$ → $I_2$, downhill by 167 kJ·mol$^{-1}$, involves for $I_2$ two new RCP within moieties {B(3)H(11)H(18)B(5)} and {B(4)B(5)H(10)B(6)}, the latter very close to the BCP between B(4) and B(5), a clear indication of ring collapse. It is noteworthy that in this reaction step there is no bond breaking, but rather bond formations with new gradient lines as follows: B(5)-B(6) and B(5)-H(11)-B(3). In the $I_2$ → $TS_3$ step, with an energy barrier of 110 kJ·mol$^{-1}$, the central hexagonal frame is more evident in $TS_3$ with three RCP, the central one and the same two satellite RCP forming the diborane moieties as in $I_2$. The former RCP close to the BCP between B(4) and B(5) in $I_2$ has now collapsed, with a B(4)-B(5) bond breaking in $TS_3$. In the final $TS_3$ → P step, with a drop of 114 kJ·mol$^{-1}$, a new RCP turns up within the {B(4)H(10)H(16)B(6)} moiety

involving the new gradient line B(4)-H(16)-B(6), with the new B(4)-H(16) bond, leading to the final D$_{3h}$ structure.

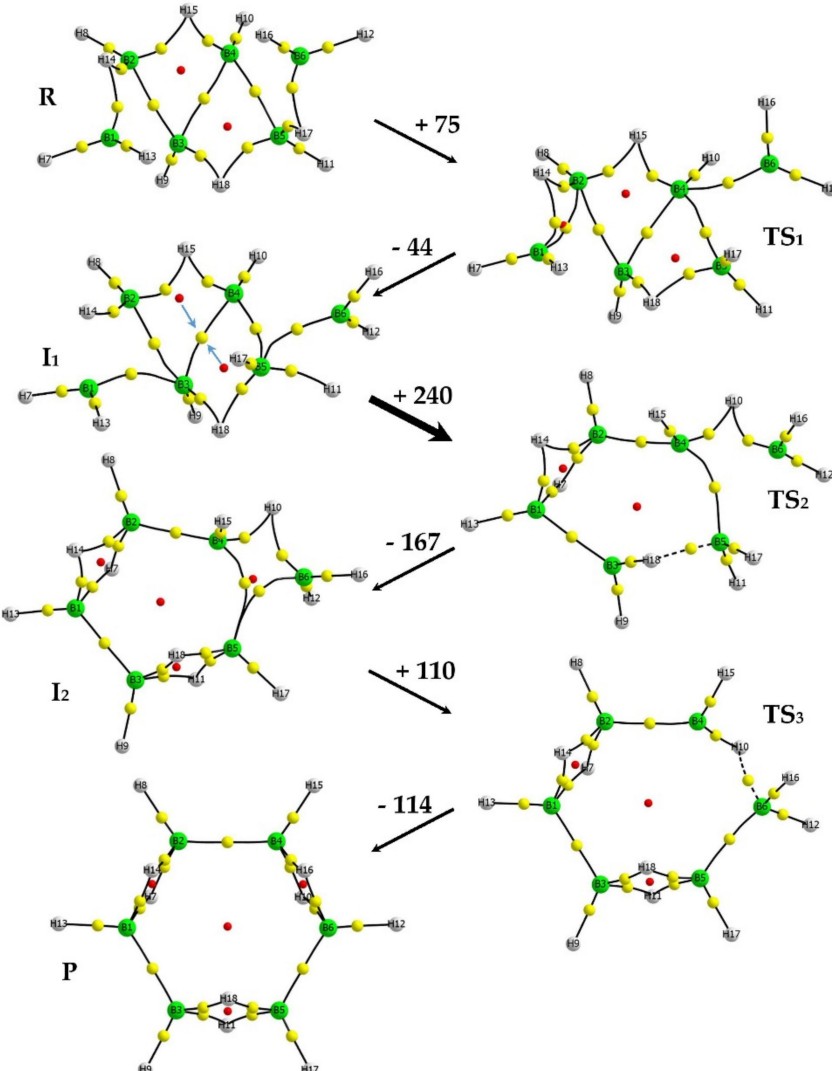

**Figure 5.** QTAIM critical points and bond paths (solid black line) of the stationary points (SP) along the IRC from Figure 2 for the 3D → 2D isomerization of hexaborane(12), with energy barriers separating the SP. Red and yellow circles correspond to ring critical points (RCP) and bond critical points (BCP), respectively. The two arrows in I$_1$ indicate the collapsing of the two RCP into one RCP in TS2. M06-2X/aug-cc-pVDZ computations.

## 4. Discussion

Reactant and Product in the 3D → 2D isomerization process of hexaborane(12) resemble, at first, two valence isomers [28]; namely, they can be transformed into each other through some reaction mechanism. In organic chemistry valence isomers are connected through pericyclic reactions [29]. The organic chemistry analogue for our case in point here could be the 3D → 2D valence isomerization of prismane to benzene [30,31]. Other benzene valence isomers have also a 3D structure, like benzvalene [32]. A quantum-chemical computation at the same level of theory shows that benzvalene and prismane lie 290 kJ·mol$^{-1}$ and 455 kJ·mol$^{-1}$ above in energy respectively, as compared to benzene, and therefore the borane isomerization is inverse, with the 2D system lying higher in energy than the 3D molecule. However, the problem here seems to be more complex: the stiffness of the central boron rhombus in R is evident given the large energy barriers involved in the isomerization process, especially the I$_1$ → TS$_2$ step, with an energy barrier of 240 kJ·mol$^{-1}$.

This barrier increases to 270 kJ·mol$^{-1}$ if we consider the energy difference of TS$_2$ with respect to R. Similar energy barriers can be found in organoboron chemistry, such as in the isomerization of borirane BC$_2$H$_5$—a triangular cyclic structure isoelectronic with the cyclopropyl cation [33]—to methyl methylideneborane, as shown in Equation (1) below. In this isomerization the computed energy barrier is 250 kJ·mol$^{-1}$, computed with highly correlated methods [33], and involves a C-C bond breaking and a hydrogen shift from B-H to one CH$_2$ group in borirane.

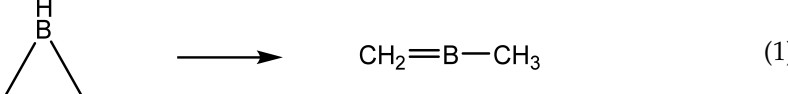

$$CH_2 \!=\! B \!-\! CH_3 \qquad (1)$$

Hydrogens in boranes play an interesting role since they provide the missing 2p electron in boron in order to resemble carbon, but an electron is ~2000 times lighter than hydrogen and the Coulombic force of the additional proton, and a boron nucleus instead of a carbon nucleus also has to be taken into account; hence the 3D diversity of borane shapes with closo, nido, arachno, and hypho structures, depending on the number of loss vertices in borane closed (closo) polyhedra [34]. The 3D → 2D isomerization of hexaborane(12) can be simplified in a 2D model of parallelogram → hexagon, as displayed in Figure 6. A 2D projected C$_2$ hexaborane(12) forms a parallelogram which transforms into a hexagon, where atoms B(1), B(2), and B(4) remain at the same point, but atoms B(3), B(5), and B(6) are shifted down along the line defined by B(3) and B(4), the two boron atoms which hold rigid the central boron rhombus in existing hexaborane(12).

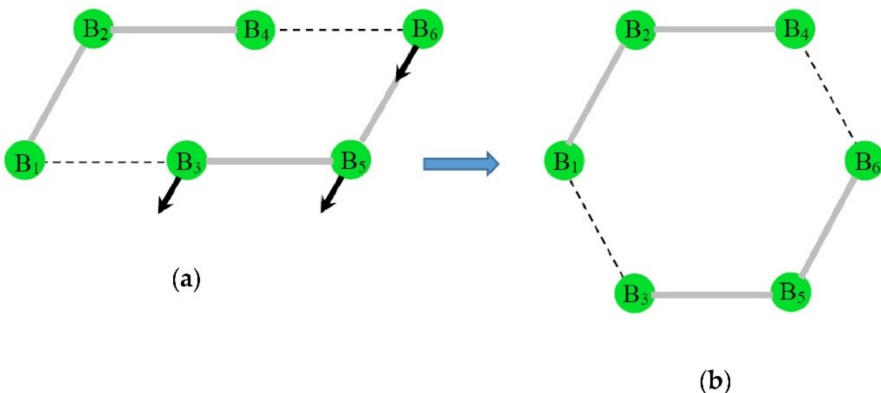

(a)

(b)

**Figure 6.** Simplified representation of the 3D → 2D isomerization of hexaborane(12): (**a**) 2D projection of the boron frame in reactant (R) C$_2$ hexaborane(12), (**b**) boron frame in 2D product (P) D$_{3h}$ hexaborane(12). The black arrows indicate the displacement of the B(3)B(5)B(6) moiety down parallel along the B(3)-B(4) line.

The above model can give a hint of the reaction mechanisms undergoing borane cage rearrangements, and further, the equivalent weak B(1)-B(3) and B(4)-B(6) interactions in the reactant would facilitate this mechanism. Recently, we reported the reaction mechanism of the photochemical and thermal isomerization between two isomers of B$_{20}$H$_{18}^{(2-)}$ [35], involving Lipscomb's diamond-square-diamond (DSD) mechanism [36], with a thermal barrier of 193 kJ·mol$^{-1}$. Although this amount of energy is still far from the 240 kJ·mol$^{-1}$ barrier needed for the 3D → 2D isomerization of hexaborane(12), it gives us a clue that large energy barriers can be surpassed in boron chemical reaction mechanisms. Indeed, the DSD mechanism is useful for predicting 3D → 3D isomerizations, but in our particular case the planarization of a 3D borane structure is, by no means, straightforward. Thus, major changes in the geometrical parameters of 3D B$_6$H$_{12}$ involve the central tilted {B(2), B(3), B(4), B(5)} rhombus, as shown in Figure 3. Other boron-boron distances undergo minor changes along the reaction mechanism. Clearly, in the I$_1$ → TS$_2$ step, two RCP in the central rhombus collapse toward the BCP between B(3) and B(4), creating a single RCP

and expanding the rhombus, as in the DSD mechanism, but with no returning to a 3D borane shape. On the other hand, the calculation of the electronic volumes in the SP along the IRC—defined as the volume inside a contour of 0.0067 e/$\text{Å}^3$ density—gathered in Table 2, provides information on the shape changes in the molecule. The molecular volume provides information on the extent of the electron density distribution of a molecule using an isodensity value. The comparison of the molecular volumes in a reaction coordinate provides some clues of the contraction or expansion of the electronic cloud along the chemical process. Thus, in the first step R $\rightarrow$ $TS_1$, the molecular volume expands by 15 $\text{Å}^3$ and the B(3)-B(4) distance shortens by 0.07 Å. In the next step $TS_1 \rightarrow I_1$, there is a shrinking of 8 $\text{Å}^3$ in the volume and a shortening of 0.04 Å in the B(3)-B(4) bond. In the limiting reaction step, $I_1 \rightarrow TS_2$, there is surprisingly a further volume shrinkage of the molecule, even lower than in R, but a striking increase of the B(3)-B(4) distance, 0.67 Å, which implies the breaking of this bond. From the $TS_2$ point onwards the volume and B(3)-B(4) distance increase up to the final product P, in the latter with a volume swell of 27 $\text{Å}^3$ as compared to R.

**Table 2.** Electronic volumes ($\text{Å}^3$) and B(3)-B(4) distance (Å) of the stationary points (SP) along the IRC for the 3D $\rightarrow$ 2D isomerization of $B_6H_{12}$—Figure 2.

| SP | Volume | B(3)-B(4) |
|---|---|---|
| R | 146.7 | 1.788 |
| $TS_1$ | 160.5 | 1.720 |
| $I_1$ | 152.5 | 1.763 |
| $TS_2$ | 145.2 | 2.434 |
| $I_2$ | 161.5 | 2.822 |
| $TS_3$ | 168.6 | 3.385 |
| P | 173.3 | 3.512 |

## 5. Conclusions

The goal of this work was to study the possibility of transforming 3D polyhedral boranes, with the particular example of existing hexaborane(12), into 2D planar borane molecules. The fact that planar $D_{3h}$ hexaborane(12) resembles benzene structurally and electronically and that 3D hexaborane(12) exists led us to this study: the 3D $\rightarrow$ 2D isomerization of hexaborane(12). By means of quantum-chemical computations we have been able to connect, through three transition states and two intermediates, the 3D and 2D structures. Along the reaction path, the most energetic step from one intermediate to a transition state involves a 240 kJ·$\text{mol}^{-1}$ energy barrier, which corresponds to expansion of the central rhombus in $B_6H_{12}$ and breaking of two boron-boron bonds. This is a large amount of energy when compared to organic chemical reactions and the proposed reaction mechanism of the 3D $\rightarrow$ 2D isomerization of hexaborane(12) throws some light on the intricacies of boron chemistry reaction mechanisms, as in the recently revisited isomerization of $B_{20}H_{18}^{(2-)}$ [35]. Thermodynamic and kinetic aspects are of paramount importance in every chemical reaction and therefore further reaction mechanisms must be studied within borane chemistry in order to understand how structures transform into one another. Finally, we should emphasize that the reaction mechanism exposed in this work is purely theoretical. Reaction mechanisms in boron chemistry are scarce, as opposed to organic chemistry, with thousands of named reactions and very well determined reaction mechanisms. The first problem in the 3D $\rightarrow$ 2D isomerization of hexaborane(12) is that the energy difference between reactant and product is very large from a thermochemical point of view. However, as mentioned above, the organic chemistry "analogue" of our case in point is somehow inverted: the very stable 2D benzene can be transformed into 3D benzvalene, lying much higher in energy, even higher than the energy difference between P and R in hexaborane(12). Perhaps using photochemical processes one could surpass the large barrier separating R and P in the 3D $\rightarrow$ 2D isomerization process of $B_6H_{12}$. The

determination of reliable reaction mechanisms in boranes is by no means trivial, but we hope that this work can throw some light on the research field of boron chemistry.

**Supplementary Materials:** The following are available online at https://www.mdpi.com/2624-8549/3/1/3/s1, Table S1: The four *styx* isomers of hexaborane(12), $B_6H_{12}$, with their structures, Lipscomb's valence structures, relative energies $\Delta E(kJ \cdot mol^{-1})$ referred to lowest energy isomer R (*styx* 4212), and the labels according to stationary points (SP) in the main text, Table S2: Selected B-B distances (Å) for the stationary points from Figure 2 of the main text, Table S3: Selected B-H distances (Å) for the stationary points from Figure 2 of the main text, Tables S4–S11: Cartesian coordinates (Å) for the optimized geometries of the stationary points considered in this work, displayed in Figure 2 of the main text, and of *styx* isomer 3303.

**Author Contributions:** Conceptualization, methodology, and supervision, J.M.O.-E., I.A., and J.E.; writing—review and editing, J.M.O.-E., I.A., J.E., M.F., and J.I.B. All authors have read and agreed to the published version of the manuscript.

**Funding:** This research was funded by Spanish MICINN, grant number CTQ2018-094644-B-C22 and Comunidad de Madrid, grant number P2018/EMT-4329 AIRTEC-CM.

**Data Availability Statement:** The data presented in this study are available in the Supplementary Materials.

**Acknowledgments:** We are grateful to Miquel Solà (University of Girona) for pointing us the first proposal of planar $D_{3h}$ hexaborane(12). This research was funded by Spanish MICINN, grant number CTQ2018-094644-B-C22 and Comunidad de Madrid, grant number P2018/EMT-4329 AIRTEC-CM.

**Conflicts of Interest:** The authors declare no conflict of interest.

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
