# Peer review of "On the 3D → 2D Isomerization of Hexaborane(12)"

_chemistry, doi:10.3390/chemistry3010003_

Round 1
Reviewer 1 Report
The authors describe a computational study on the reaction mechanism of 3D hexaborane planarization, identifying the potential reaction path, along with the transition states and the intermediates of the process.
The work provides an insight into a still purely theoretical transformation. The paper can be suitable for publication, after some amendments in the text and the images, and an enrichment in the results discussion
In the following some suggestions:
- A suggestion: Figure 1 is helpful in understanding the structures however perhaps the ball and stick model does not make it easy to visualize the 3D structures. Perhaps a tube model could provide an easier visualization. This is also true in Fig.2 for better visualizing the distortions.
- Introduction
- The reaction under study should be better put into context. Why is this reaction of interest, although the product is “unknown”? - What is the connectivity rule?
- methods
- Line 56: I would replace model chemistry with method or level of theory.
- please briefly explain what physical insight provides the determination of molecular volume.
- Results.
- Figure 2 and text: personally, I would choose kcal/mol as measurement unit for the energy, being more common in the context of chemical reactivity.
- Figure 3 and 4. I would avoid yellow as a line colour.
- At line 86, R isomer stability is mentioned but without further information (example structural description, other relevant isomers, quantification of stability energy). Please clarify and substantiate with data (perhaps in SI).
- In my opinion Figure 2 is not adequately commented from a “chemical” perspective. For example, a comment on the barrier heights, on which steps are expected to be determinant, etc.
- Personally, I found the description of the structural characteristics (bond lengths) a bit lengthy. Perhaps I would make it a bit shorter and more focused in highlighting the most relevant geometrical distortions in the transformation steps, also drawing a relation with the energetic profile and barriers.
- please explain the meaning of the ring critical points descriptor, and it physical significance.
4.discussion
- Equation (1) is not visible in the text.
-the discussion contains several interesting aspects that however in my view are sometimes disconnected. For example: it is pointed the relevance of hydrogens in boranes, however it is not clear the relation with the present study; on what basis the 3D->2D isomerization can be boiled down to a 2D model of parallelogram->hexagon.
-A comment on the expected experimental feasibility of the reaction, in my view would be interesting
- Conclusion
The conclusions are somehow poor, not adequately summarizing all the information about the reaction that can be inferred from the study. I would recommend to extend.
Author Response
Please find attached the reply in file.

Reviewer 2 Report
This is an important contribution to understanding the relationship between boranes and hydrocarbons by focussing on the borane analogue of benzene, namely B6H12. Thus thus work, unlike the previous work cited by the authors in references 7 and 8, focuses on the mechanism of conversion of the non-planar experimental structure of B6H12 to a high-energy planar isomer isoelectronic with the planar benzene. The theoretical methods used in this work are sound well-established methods for such systems and are adequately described.
Publication of this interesting paper in Chemistry is recommended even though the only connection with experiment is the fact that the "reactant" B6H12 is a known compound.
On reading the manuscript I noted the following minor points that should be corrected before publication:
• The boranes B10H14 and B18H22 are actually air-stable white solids which can be handled under ambient conditions so the statement on line 23 of the Introduction is actually misleading.
• The word is "hypho" rather than "hypo" when referring to the structures of the 3D boranes (line 230)
Author Response
Please find attached file with replies to Reviewer 2
